# Efficacy and Safety of Pathogen-Reduced Platelets Compared with Standard Apheresis Platelets: A Systematic Review of RCTs

**DOI:** 10.3390/pathogens11060639

**Published:** 2022-06-01

**Authors:** Ilaria Pati, Francesca Masiello, Simonetta Pupella, Mario Cruciani, Vincenzo De Angelis

**Affiliations:** National Blood Centre, Italian National Institute of Health, 00161 Rome, Italy; francesca.masiello@iss.it (F.M.); simonetta.pupella@iss.it (S.P.); crucianimario@virgilio.it (M.C.); vincenzo.deangelis@iss.it (V.D.A.)

**Keywords:** pathogen reduction technology, pathogen inactivation, pathogen-reduced platelets, bleeding, adverse events, platelet count increment, refractoriness, alloimmunization, systematic review

## Abstract

In this systematic review, we evaluate the efficacy and safety of blood components treated with pathogen reduction technologies (PRTs). We searched the Medline, Embase, Scopus, Ovid, and Cochrane Library to identify RCTs evaluating PRTs. Risk of bias assessment and the Mantel–Haenszel method for data synthesis were used. We included in this review 19 RCTs evaluating 4332 patients (mostly oncohematological patients) receiving blood components treated with three different PRTs. Compared with standard platelets (St-PLTs), the treatment with pathogen-reduced platelets (PR-PLTs) does not increase the occurrence of bleeding events, although a slight increase in the occurrence of severe bleeding events was observed in the overall comparison. No between-groups difference in the occurrence of serious adverse events was observed. PR-PLT recipients had a lower 1 and 24 h CI and CCI. The number of patients with platelet refractoriness and alloimmunization was significantly higher in PR-PLT recipients compared with St-PLT recipients. PR-PLT recipients had a higher number of platelet and RBC transfusions compared with St-PLT recipients, with a shorter transfusion time interval. The quality of evidence for these outcomes was from moderate to high. Blood components treated with PRTs are not implicated in serious adverse events, and PR-PLTs do not have a major effect on the increase in bleeding events. However, treatment with PRTs may require a greater number of transfusions in shorter time intervals and may be implicated in an increase in platelet refractoriness and alloimmunization.

## 1. Introduction

In recent years, much progress in ensuring the safety of blood and blood components has been observed, especially in order to reduce the risk of transmission of infections.

In Italy, donations are routinely screened for known virus infections (hepatitis C virus (HCV), human immunodeficiency virus (HIV), hepatitis B virus (HBV), and *Treponema pallidum*) [1]. However, there remains a residual risk of transfusion transmission both for known pathogens and for emerging pathogens for which blood donations are not routinely tested [2,3,4].

In addition, the safety of blood and blood components has also improved with the introduction of measures to reduce the risk of bacterial contamination (diversion of the first 10 mL of blood, accurate skin disinfection at the venipuncture site, and the adoption of standardized and validated operating procedures for blood processing) [1,5].

However, cases of transfusion-associated bacterial sepsis (TABS) from contaminated blood components have been described [6,7]. The transmission of the infection to the recipient is directly related to the amount of blood component transfused, the pathogen concentration, and the degree of immunocompetence of the transfused patient [8,9].

Plasma and cryoprecipitate, stored in the frozen state, are rarely associated with bacterial contamination phenomena. Although in the literature cases of contamination by Gram-negative pathogens of plasma and cryoprecipitate during thawing in contaminated thermostatic water baths are reported, the risk of TABS is, overall, almost nil [10].

On the contrary, platelets appear to be the blood component most involved in TABS due to the storage temperature between 20 and 24 °C, a condition favorable to the growth of a wide spectrum of bacteria [10]. Most platelet concentrates are transfused to immunosuppressed patients (hematological, oncohematological patients, and hematopoietic stem cell transplant recipients), who frequently have a state of pyrexia associated with the underlying disease; therefore, TABS may not be recognized promptly [11].

A survey conducted by the Italian National Blood Centre in 2019 aimed at evaluating the procedures in use at the BEs for the microbiological sterility controls of blood components found that about 87% of the responding BEs (129 out of a total of 278 BEs) perform microbiological control using automated or semiautomated culture systems. Platelets resulted the blood components most involved in bacterial contamination.

According to the European Centre for Disease Prevention and Control (ECDC), between 2010 and 2016, 1% of notifications of serious adverse reactions in recipients concerned transfusion-transmitted infections (TTIs); 63% of these infections caused by bacterial contamination; platelets from apheresis are the blood component most involved in TABS [12].

In 2019, US Food and Drug Administration (FDA) data on “Fatalities” after transfusion reported one case of a bacterial-contamination-related fatality, with an imputability probable, attributed to a pooled platelet transfusion [13].

The pathogen reduction technologies (PRTs) were introduced with the aim of reducing the risk of transfusion-transmitted infections [14].

PRTs have nucleic acids as their primary target, and therefore, in plasma and platelets, they can inactivate viruses, bacteria, and parasites [14].

The PRTs currently available for platelet inactivation are the Intercept^®^ Blood System (IBS), which uses a synthetic psoralen, amotosalen HCl, and low-energy ultraviolet light (ultraviolet A (UVA)) as the active compound; the Mirasol^®^ system, which uses riboflavin (vitamin B2), associated with irradiation with UVB (ultraviolet B) light; and the Theraflex^®^ UV-Platelets system, which uses UVC (ultraviolet C) light [14].

The application of PRTs to platelet concentrates allows us to extend the shelf life of platelets from 5 to 7 days [15]. However, the results of platelet efficacy studies after inactivation are highly controversial. Some studies report that the PRTs can negatively impact platelet vitality and hemostatic function [16,17]; others, on the contrary, reaffirm platelet efficacy and safety after inactivation treatment compared with products not photochemically treated, and do not report an increase in adverse reactions in treated patients [17,18,19,20,21,22,23,24,25,26,27,28,29,30]; and other studies do not provide useful information [31,32,33,34].

A number of randomized controlled clinical trials (RCTs) have evaluated the treatment of blood components with PRTs, and systematic reviews and meta-analysis have been subsequently published [35,36]. Since then, however, new RCTs have been published that provide an accurate and updated summary of the best available evidence; therefore, we undertook a new systematic review on the efficacy and safety of pathogen-reduced platelets (PR-PLTs) compared with standard apheresis platelets (St-PLTs).

## 2. Results

The search identified a total of 776 potentially relevant records. After removal of duplicates, 523 records remained, of which 480 were excluded on the basis of the abstract and/or title. The search identified 43 records that appeared relevant on the basis of their full text or abstract using the original inclusion/exclusion criteria (Figure 1).

Twenty-four of them were excluded (reviews, protocols of RCTs, nonrandomized studies, duplicates, studies containing no informative data). Nineteen RCTs were included in the systematic review (see Table 1 for the main characteristics and results of the included studies).

Overall, 4606 patients were enrolled in the 19 RCTs selected for the review. Of the 19 trials included in the systematic review, 10 compared Intercept^®^ PR-PLTs with St-PLTs [16,18,19,20,22,23,25,30,32,34], 6 Mirasol^®^ PR-PLTs with St-PLTs [17,26,27,28,29,31], 2 subgroups of patients receiving either Intercept^®^ or Mirasol^®^ PR-PLTs with St-PLTs [21,33], and 1 Theraflex^®^ PR-PLTs with St-PLTs [24] (Table 1).

In 17 studies, PR-PLTs was compared with St-PLTs. Two other studies considered the treatment of whole blood and RBCs with PRT, respectively.

Fifteen were parallel-group RCTs, and 4 were randomized crossover trials. Of the 4606 patients enrolled in the trials, 4332 received at least 1 platelet transfusion (2613 in Intercept^®^ platelet trials, 1299 in Mirasol^®^ platelet trials, and 171 Theraflex^®^ platelet trials).

One trial included children requiring cardiac surgery (16 participants) or adults requiring a liver transplant (28 participants). All of the other participants were thrombocytopenic patients who had a hematological or oncological diagnosis.

With the exception of one trial from Ghana, studies were conducted in industrialized countries, including the USA, Canada, and Europe (France, Germany, Belgium, the Netherlands, Sweden, Spain, and Italy).

### 2.1. Risk of Bias in Included Studies

Fourteen out of 19 reports were at high or unclear risk of bias for 1 or more domains; 6 were at high risk of bias in 1 domain, and 10 were at unclear risk of bias in 1 or more domains (Figure 2).

Five reports [18,20,21,28] were at low risk of bias in all the domains. We assessed 5 studies as being at unclear risk of selection bias because they were unclear about the random sequence generation and the allocation concealment, while 14 studies were at low risk of selection biases. There were 4 open-label trials, and they were graded as high risk of performance bias (blinding of participants and personnel). Six studies were graded at unclear risk of detection bias due to the fact that they did not provide information to permit judgement about “high” or “low” risk of bias related to the blinding of participants and personnel. Nine studies were reported as double blind. Fourteen studies (73.6%) were graded at low risk of detection bias due to the fact that the assessor was blinded to treatment allocation, whereas 5 studies were graded at unclear risk of detection bias because they did not provide information to permit judgement about “high” or “low” risk of bias related to the blinding of outcome assessors. One trial [33] was judged at high risk of attrition bias because only 179 out of the 237 patients enrolled in the Intercept^®^ arm of the original study [21] and 179 out of 201 of the Mirasol^®^ arm were included in this analysis. Other 5 studies were judged at unclear risk of attrition bias, and the remaining 13 (68%) studies were judged at low risk of bias. Risk of bias for selective reporting or other potential source of bias was present in 3 studies (high risk in 1 case, unclear risk in 2), 2 of which available only as abstracts [32,34].

### 2.2. Effects of Interventions

See Appendix A, data and analyses.

### 2.3. Outcomes

The outcomes reported (Table 2) were bleeding events, adverse events, number of patients with acute transfusion reactions, platelet count increment (CI) and corrected count increment (CCI), number of patients with platelet refractoriness and alloimmunization, number of platelet transfusions/patient, and number of RBC transfusions/patient.

#### 2.3.1. Bleeding Events

Bleeding events at more than 7 days were reported as any bleeding events (WHO grades 1 to 4), clinically significant bleeding (WHO grade ≥ 2), and severe bleeding (WHO grade ≥ 3) (Figure 3).

“Any bleeding” was reported in 7 trials (1931 patients). No difference in the occurrence of bleeding was found in the overall analysis (RR 1.03; 95% *CIs*, 0.85/1.24; *p* = 0.79), in a subgroup analysis of 6 Intercept^®^ trials ((RR 0.99; 95% *CIs*, 0.80/1.22; *p* = 0.91), and in a Mirasol^®^ trial (RR 1.38; 95% *CIs*, 0.95/2.02; *p* = 0.09) (Figure 3a). No difference was found in the occurrence of “clinically significant bleeding” in a subgroup analysis of 5 Intercept^®^ trials (RR 1.05; 95% *CIs*, 0.95/1.16; *p* = 0.32) and 4 Mirasol^®^ trials (RR 1.39; 95% *CIs*, 0.99/1.96; *p* = 0.06), although in the overall analysis (9 trials, 3033 patients), a higher number of bleeding events was observed in PR-PLTs recipients compared with the control group (RR 1.16; 95% *CIs*, 1.02/1.32; *p* = 0.03) (Figure 3b). The quality of the evidence for overall bleeding and serious bleeding was graded as moderate due to heterogeneity. The occurrence of “severe bleeding” was comparable between groups, in the overall analysis (11 trials; 3297 patients; RR 1.09; 95% *CIs*, 0.76/1.56; *p* = 0.65), and in a subgroup analysis of 6 Intercept^®^ trials (RR 1.15; 95% *CIs*, 0.62/2.13; *p* = 0.66), in 4 Mirasol^®^ trials (RR 1.34; 95% *CIs*, 0.74/2.42; *p* = 0.34), and in a Theraflex^®^ trial (RR 0.32; 95% *CIs*, 0.01/7.79; *p* = 0.49) (Figure 3c). For the outcome severe bleeding, the quality of the evidence was graded as high (Table 2).

#### 2.3.2. Adverse Events

Adverse events complicating platelet transfusion were reported as any adverse events and/or serious adverse events, as defined in individual studies, occurring during the study and follow-up periods (Figure 4).

Several studies reported also acute mild transfusion reactions (e.g., rigors, fever, skin rash, and urticaria). Any adverse event was reported in 11 trials (12 records, 3445 patients). No between-groups difference in the occurrence of any adverse event was found in the 6 Intercept^®^ trials (RR 1.01; 95% *CIs*, 0.89/1.15; *p* = 0.84). By contrast, a higher incidence of adverse events in the PR-PLTs compared with the St-PLTs was found in the overall analysis (RR 1.09; 95% *CIs*, 1.01/1.19; *p* = 0.03) and in a subgroup analysis of Mirasol^®^ trials ((RR 1.16; 95% *CIs*, 1.04/1.30; *p* = 0.006). The quality of the evidence was graded as low due to heterogeneity and risk of bias. Serious adverse events were reported in 13 studies (3247 patients), 7 with Intercept^®^ (2078 patients), 5 with Mirasol^®^ (1027 patients), and 1 with Theraflex^®^ (142 patients). No between-groups differences in the occurrence of a serious adverse event was found in the overall analysis (RR 1.01; 95% *CIs*, 0.82/1.24) and in a subgroup analysis of Intercept^®^ trials (RR 1.08; 95% *CIs*, 0.86/1.37), a Mirasol^®^ trial (RR 0.78; 95% *CIs,* 0.49/1.25), and a Theraflex^®^ trial (RR not estimable because no events were reported in either group): moderate quality of evidence due to imprecision because most of the trials were underpowered to detect the occurrence of rare outcomes. Acute transfusion reactions, as defined among the adverse events or the serious adverse events (e.g., rigors, fever, skin rash, and urticaria) were reported in 7 trials, and the occurrence did not differ between groups (RR 0.95; 95% *CIs*, 0.62/1.47).

#### 2.3.3. Platelet Count Increment (CI) and Corrected Count Increment (CCI) at 1 and 24 h

The outcome 1 h CI was reported in 10 reports (8 Intercept^®^, 1 Mirasol^®^, and 1 Theraflex^®^) and 1847 patients; 1 h CCI in 11 reports (8 Intercept^®^, 2 Mirasol^®^, 1 Theraflex^®^) and 1933 patients; 24 h CI in 9 reports (7 Intercept^®^, 1 Mirasol^®^, 1 Theraflex^®^) and 1800 patients; 24 h CCI in 11 reports (8 Intercept^®^, 2 Mirasol^®^, 1 Theraflex^®^) and 2435 patients (Figure 5 and Figure 6).

Patients who received PRT had a significantly lower 1 h CI in the overall analysis (MD −7.10; 95% *CI*, −10.58/−3.62; *p* < 0.00001), in the 8 Intercept^®^ reports (MD −7.24; 95% CI, −11.4/−3.06; *p* = 0.0007), and in a Theraflex^®^ study (MD −5.01; 95% *CI*, −8.55/−1.47; *p* = 0.006); in the 2 Mirasol^®^ reports, there was a trend towards a lower 1 h CI, but the difference was not statistically significant (MD −8.90; 95% *CI*, −18.47/0.67; *p* = 0.07). Patients who received PRT had a significantly lower 1 h CCI in the overall analysis (MD −3.15; 95% *CI*, −4.29/−2.0; *p* < 0.00001), in the 8 Intercept^®^ reports (MD −2.97; 95% *CI*, −4.47/−1.48; *p* < 0.0001), in the 2 Mirasol^®^ reports (MD −4.12; 95% *CI*, −6.29/−1.96; *p* = 0.0002), and in a Theraflex^®^ study (MD −2.63; 95% *CI*, −4.44/−0.82; *p* = 0.004).

Likewise, patients who received PR-PLTs had a significantly lower 24 h CI in the overall analysis (MD −6.65; 95% *CI*, −8.44/−4.86; *p* < 0.00001), in the 7 Intercept^®^ reports (MD −7.61; 95% *CI*, −9.45/−5.77; *p* < 0.00001), in 1 Mirasol^®^ report (MD −4.30; 95% *CI*, −7.38/−1.22; *p* = 0.006), and in a Theraflex^®^ study (MD −3.84; 95% *CI*, −7.06/−0.62; *p* = 0.02), as well as a significantly lower 24 h CCI in the overall analysis (MD −3.18; 95% *CI*, −3.96/−2.41; *p* < 0.00001), in the 8 Intercept^®^ reports (MD −3.51; 95% *CI*, −4.44/−2.58; *p* < 0.00001), in the 2 Mirasol^®^ reports (MD −2.37; 95% *CI*, −3.68/−1.06; *p* = 0.0004), and in a Theraflex^®^ study (MD −2.08; 95% *CI*, −3.84/−0.32; *p* = 0.02). For the CI and CCI outcomes, the quality of the evidence was graded as moderate due to inconsistency (between-trials heterogeneity).

#### 2.3.4. Platelet Refractoriness and Platelet Alloimmunization

There was heterogeneity in the definition of refractoriness (e.g., 2 successive 1 or 24 h CCIs below 7.5 × 10^3^ or 4.5/5 × 10^3^), and in some of the selected trials, subjects with a previous history of clinical refractoriness to platelet transfusions were not eligible for inclusion in the analysis. Ten reports (2380 participants) reported the number of patients experiencing platelet refractoriness. Participants who received pathogen-reduced platelet transfusions had an increased risk of developing platelet refractoriness in the overall analysis (RR 2.59: 95% *CIs*, 1.98/3.39; *p* < 0.00001) and in a subgroup analysis of 6 Intercept^®^ trials (RR 2.85; 95% *CIs*, 1.96/4.15; *p* < 0.00001) and 3 Mirasol^®^ trials (RR 2.46; 95% *CIs*, 1.61/3.76; *p* < 0.0001); in the Theraflex^®^ trial, there was a trend towards a higher refractoriness in PR-PLT recipients compared with St-PLTs, but the difference was not statistically significant (RR 1.81; 95% *CI*, 0.71/4.64; *p* = 0.22) (Figure 7). The quality of the evidence was graded as high.

Platelet refractoriness and refractoriness specifically due to alloimmunization were reported in 11 reports (2628 participants) (Figure 8). Similarly to the refractoriness analysis, patients who received PR-PLT transfusions had an increased risk of developing platelet refractoriness and alloimmunization in the overall analysis (RR 1.77; 95% *CIs*, 1.47/2.13; *p* < 0.00001) and in a subgroup analysis of 7 Intercept^®^ trials (RR 1.61; 95% *CIs*, 1.28/2.02; *p* < 0.0001) and of 3 Mirasol^®^ trials (RR 2.14; 95% *CIs*, 1.50/3.07; *p* < 0.0001); in the Theraflex^®^ trial, there was a trend towards a higher refractoriness in PR-PLTrecipients compared with St-PLTs, but the difference was not statistically significant (RR 1.77; 95% *CI*, 0.74/4.24; *p* = 0.20). The quality of the evidence was graded as high.

#### 2.3.5. Platelet Transfusions, Platelet Transfusion Interval, and Red Blood Cell Transfusions

The number of platelet transfusions/patient and the number of RBC transfusions/patient were reported in 9 trials (2194 and 2193 participants, respectively) (Figure 9). Patients who received PR-PLT transfusions required more platelet transfusions in the overall analysis (MD 1.04; 95% *CI*, 0.84/1.24; *p* < 0.00001), and in a subgroup analysis of 6 Intercept^®^ trials (MD 1.07; 95% *CI*, 0.85/1.29; *p* < 0.00001), 2 Mirasol^®^ trials (MD 1.06; 95% *CI*, 0.26/1.87; *p* = 0.009), and the Theraflex^®^ trial (MD 0.73; 95% *CI*, 0.04/1.42; *p* = 0.04). As far as RBC transfusion is concerned, patients who received PR-PLT transfusions required more RBC transfusions in the overall analysis (MD 0.32; 95% *CI*, 0.14/0.50; *p* = 0.0004) and in a subgroup analysis of 6 Intercept^®^ trials (MD 0.27; 95% *CI*, 0.08/0.47; *p* = 0.006) and the Theraflex^®^ trial (MD 0.73, 95% *CIs* 0.04/1.42; *p* = 0.04); in the 2 Mirasol^®^ trials, the difference was not statistically significant (MD 0.42; 95% *CI*, 0.12/0.96; *p* = 0.13). For these two outcomes, the quality of the evidence was graded as high.

Platelet transfusion interval was reported in 11 reports (2424 patients) (Figure 10). The day of the next platelet transfusion was significantly shorter in PR-PLT recipients in the overall analysis (RD −0.22; 95% *CIs*, −0.41/−0.03; *p* = 0.02) and in the subgroup of Intercept^®^ trial (RD −0.33; 95% *CIs*, −0.57/−0.10; *p* = 0.006), but not in the subgroup of Mirasol^®^ trials (RD −0.11; 95% *CIs*, −0.30/0.08; *p* = 0.25); in the Theraflex^®^ trial, the interval was shorter in St-PLT recipients (RD 0.51; 95% *CIs*, −0.20/1.22; *p* = 0.16), but the difference was not statistically significant. For this outcome, the quality of the evidence was graded as moderate due to inconsistency (between-trials heterogeneity and significant subgroup differences).

## 3. Discussion

The main aims of this systematic review and meta-analysis are to comprehensively evaluate the efficacy and safety of platelets treated with currently available PRTs, especially by comparing the treated products with the standards in terms of reduction of bleeding and transfusion-related adverse reactions.

We included in this review 19 RCTs evaluating 4332 patients receiving platelet transfusions treated with three different PRTs. The majority of the participants were patients with hematological malignancies. On average, compared with St-PLTs, PR-PLT transfusion does not increase the occurrence of bleeding events, although a slight increase in the occurrence of severe bleeding events was observed in the overall comparison. The quality of evidence for these outcomes was from moderate to high. No between-groups difference in the occurrence of serious adverse events was observed, in the overall analysis, and in subgroup analyses of Intercept^®^, Mirasol^®^, and Theraflex^®^ trials (moderate quality of evidence). In the overall analysis and in the subgroup of Mirasol^®^ trials, but not in the subgroup analysis of Intercept^®^ trials, overall adverse events were more commonly observed in PR-PLT recipients compared with St-PLT recipients; the quality of the evidence was graded as low due to risk of bias (differences in the definition and assessment of overall adverse events) and inconsistency (substantial heterogeneity between trials). On average, PR-PLT recipients had a lower 1 and 24 h CI and a lower 1 and 24 h CCI (moderate quality of evidence due to between-trials heterogeneity). There was also high quality of evidence that the number of patients with platelet refractoriness and the number of patients with platelet refractoriness and alloimmunization were significantly higher in PR-PLT recipients compared with St-PLT recipients. In the same way, PR-PLT recipients had a higher number of platelet transfusions and RBC transfusions compared with St-PLT recipients, with a shorter transfusion time interval.

Compared with the previously published Cochrane systematic review [36], the current review includes 7 additional trials, 1 with Intercept^®^ [25], 4 with Mirasol^®^ [17,26,28,29], 1 with both Intercept^®^ and Mirasol^®^ [33], and 1 with Theraflex^®^ [24]; and for some of the outcomes evaluated, the number of individuals included in the analyses almost doubled.

Similar to the Cochrane review [35], our review reports a significant reduction of the 1 and 24 h CI and CCI in PR-PLT recipients compared with St-PLTs recipients, as well as an increased platelet transfusion and RBC demand. However, unlike the Cochrane review, we did not observe an increase in overall bleeding events (RR 1.03; 95% *CI*, 0.85 to 1.24), but a slight increase in severe bleeding events in the overall analysis (RR 1.16; 95% *CIs*, 1.02/1.32); also, we graded the evidence about serious bleeding events as high level of certainty, compared with the moderately low quality found in the Estcourt et al. review [35]. Moreover, unlike the Cochrane review, we did not include all-cause mortality and attributable mortality (e.g., due to infection or bleeding) among the outcomes, because there was no evidence in the trials considered that overall mortality and attributable mortality were related to platelet transfusions, but rather to the underlying clinical conditions (most of the participants were patients with thrombocytopenia from oncological diseases).

In conclusion, the results of our review show that the treatment of blood components with PRTs is not implicated in serious adverse events in the recipient. In particular, the treatment of platelets does not have a major effect on the increase in bleeding events. However, treatment with PRTs may require a greater number of transfusions in shorter time intervals and may be implicated in an increase in platelet refractoriness and alloimmunization. To better understand and define the adverse events and any limitations related to the treatment with PRTs, it is important to conduct further investigations in this regard also through a comparative analysis of the different PRTs.

## 4. Materials and Methods

This systematic review was conducted according to recommended PRISMA checklist guidelines [37]. The protocol has been registered in PROSPERO (registration number CRD42022320422), the international prospective register of systematic reviews. The review is aimed at evaluating the safety and effectiveness of PR-PLTs in people undergoing platelet transfusions. We included RCTs comparing the transfusion of PR-PLTs with St-PLTs. Three different types of pathogen reduction technologies were considered, including the Intercept^®^, Mirasol^®^, and Theraflex^®^ systems.

### 4.1. Search Strategy

A computer-assisted literature search of the Medline (through PubMed), Embase, Scopus, Ovid, and Cochrane Library was performed (latest search in February 2022) to identify RCTs evaluating pathogen reduction technologies. A combination of the following text words was used: platelet AND pathogen reduction, pathogen reduction platelet, Mirasol^®^ platelet, Intercept^®^ platelet, Theraflex^®^ platelet. In addition, we checked the reference lists of the most relevant items (original studies and reviews) in order to identify potentially eligible studies not captured by the initial literature search. For the search, no restriction on language was applied.

### 4.2. Data Collection and Analysis

For each RCT included in the systematic review, the following data were extracted by two reviewers (MC and IP) independently: first author, year of publication, details of intervention in study and control group, sample size, pathogen reduction technology used, control group, outcome measurements, and main results. Measures of treatment effect were mean differences (MD) together with 95% confidence intervals (*CI*) for continuous outcome measures and risk ratio (RR) with 95% *CI* for dichotomous outcomes. For continuous outcomes, the score had to be reported as mean and standard deviation (SD); when studies reported other dispersion measures, such as median and range, or standard error (SE) of the mean or 95% *CI* of the mean, we calculated the mean and SD from these measures in order to perform the relevant meta-analytical pooling [38,39]. Disagreement was resolved by consensus and by the opinion of a third reviewer (FM), if necessary.

The study weight was calculated using the Mantel–Haenszel method. We assessed statistical heterogeneity using t2, Cochran’s Q, and *I*^2^ statistics. The *I*^2^ statistic describes the percentage of total variation across trials that is due to heterogeneity rather than sampling error. In the case of not important heterogeneity (*I*^2^ < 40), studies were pooled using a fixed-effects model. Where values of *I*^2^ were >40, a random-effects analysis was undertaken. All calculations were performed using Excel and RevMan 5.4.

### 4.3. Outcomes

The outcomes included in the analysis were: bleeding events (any bleeding event, significant bleeding, and serious bleeding) mainly using two bleeding scales: WHO Bleeding Scale Grades 0 to 4 and Common Terminology Criteria for Adverse Events (CTCAE) Grades 1 to 5 or equivalent; adverse events graded for clinical severity (any adverse event, serious adverse event, acute transfusion reactions); platelet CI and CCI at 1 and 24 h; number of patients with platelet refractoriness and number of patients with platelet refractoriness and alloimmunization; number of platelet transfusions/participant and number of red blood transfusions/participant; and platelet transfusion interval (day of the next platelet transfusion).

### 4.4. Subgroup Analyses

We undertook subgroup analyses according to type of PRT methodology used (Intercept^®^, Mirasol^®^, and Theraflex^®^).

### 4.5. Assessment of Risk of Bias in Included Studies

Two review authors (MC, IP) independently assessed the risk of bias of each included study following the domain-based evaluation described in the Cochrane Handbook for Systematic Reviews of Interventions [39]. They discussed any discrepancies and achieved consensus on the final assessment. The Cochrane ’risk of bias’ tool addresses six specific domains: sequence generation, allocation concealment, blinding, incomplete data, selective outcome reporting, and other issues relating to bias. We have presented our assessment of risk of bias using two ‘risk of bias’ summary figures: (1) a summary of bias for each item across all studies and (2) a cross tabulation of each trial by all of the ‘risk of bias’ items (Figure 2).

We used the principles of the GRADE system to assess the quality of the body of evidence associated with specific outcomes, and constructed a ‘summary of findings’ table (Table 2) using RevMan 5 [40].

These tables present key information concerning the certainty of the evidence, the magnitude of the effects of the interventions examined, and the sum of available data for the main outcomes [39]. The ‘summary of findings’ tables also include an overall grading of the evidence related to each of the main outcomes using the GRADE approach, which defines the certainty of a body of evidence as the extent to which one can be confident that an estimate of effect or association is close to the true quantity of specific interest. The certainty of a body of evidence involves consideration of within-trial risk of bias (methodological quality), directness of evidence, heterogeneity, precision of effect estimates, and risk of publication bias [41].

When evaluating the ‘risk of bias’ domain, we downgraded the GRADE assessment when we classified a study as being at high risk of bias for one or more of the following domains: selection, attrition, performance, detection, reporting, and other bias, or when the ‘risk of bias’ assessment for selection bias was unclear (this was classified as unclear for either the generation of the randomization sequence or the allocation concealment domain).

We have presented the following outcomes in the ‘summary of findings’ table: bleeding events, adverse events, and platelet CI and CCI at 1 and 24 h; number of patients with platelet refractoriness and number of patients with platelet refractoriness and alloimmunization; and number of platelet transfusions/participant and number of red blood transfusions/participant.

## 5. Conclusions and Perspectives

Blood components treated with PRTs are not implicated in serious adverse events, and PR-PLTs do not have a major effect on the increase in bleeding events. However, treatment with PRTs may require a greater number of transfusions in shorter time intervals and may be implicated in an increase in platelet refractoriness and alloimmunization. However, to better understand and define the clinical effectiveness and safety of PRTs, further investigations directly comparing different pathogen reduction techniques and their use in non-hemato-oncological patients are required.

## Figures and Tables

**Figure 1 pathogens-11-00639-f001:**
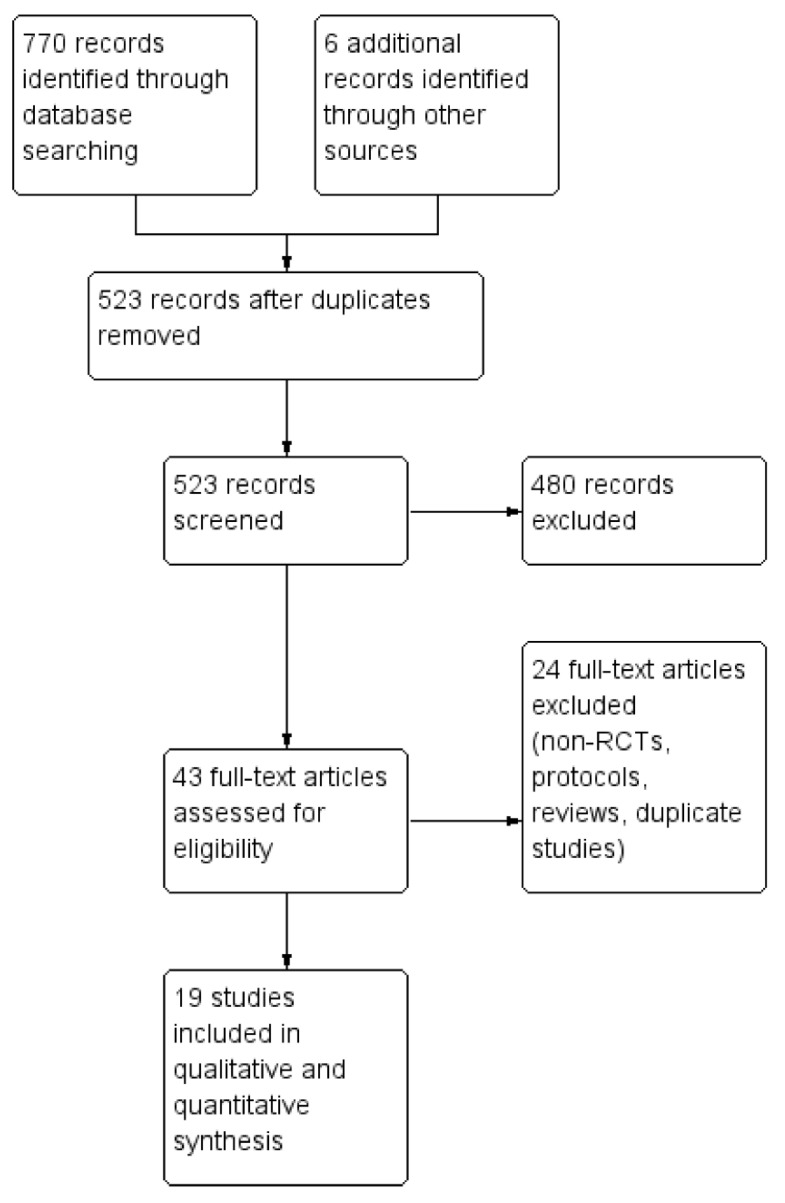
Study flow diagram. PRISMA flowchart summarizing the inclusion and exclusion of studies.

**Figure 2 pathogens-11-00639-f002:**
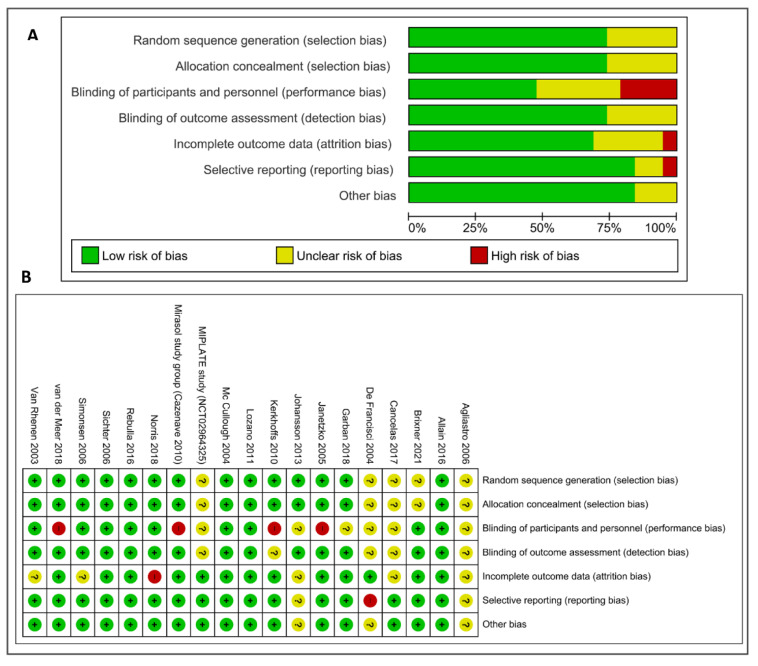
Risk of bias: (**A**) Risk of bias graph: review authors’ judgements about each risk of bias item presented as percentages across all included studies; (**B**) risk of bias summary: review authors’ judgements about each risk of bias item for each included study (see Material and Methods for details about the assessment).

**Figure 3 pathogens-11-00639-f003:**
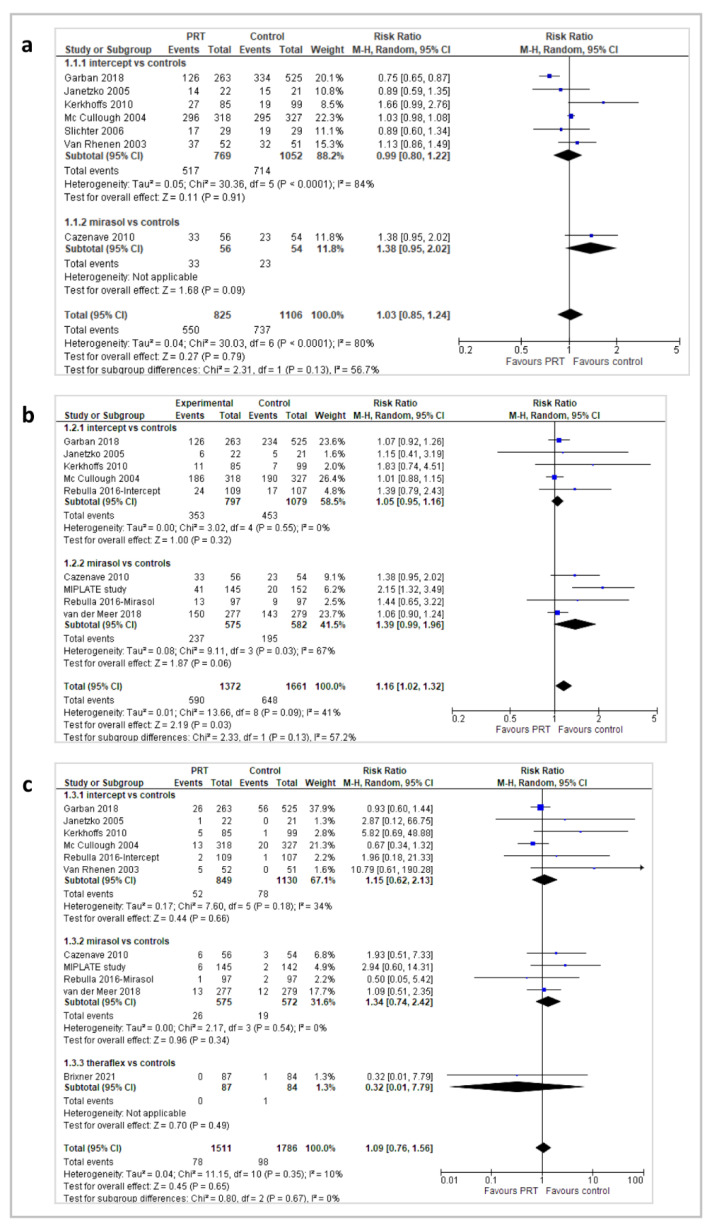
Bleeding events: (**a**) any bleeding events; (**b**) significant bleeding (grade ≥ 2) episodes; (**c**) severe (≥3) bleeding episodes.

**Figure 4 pathogens-11-00639-f004:**
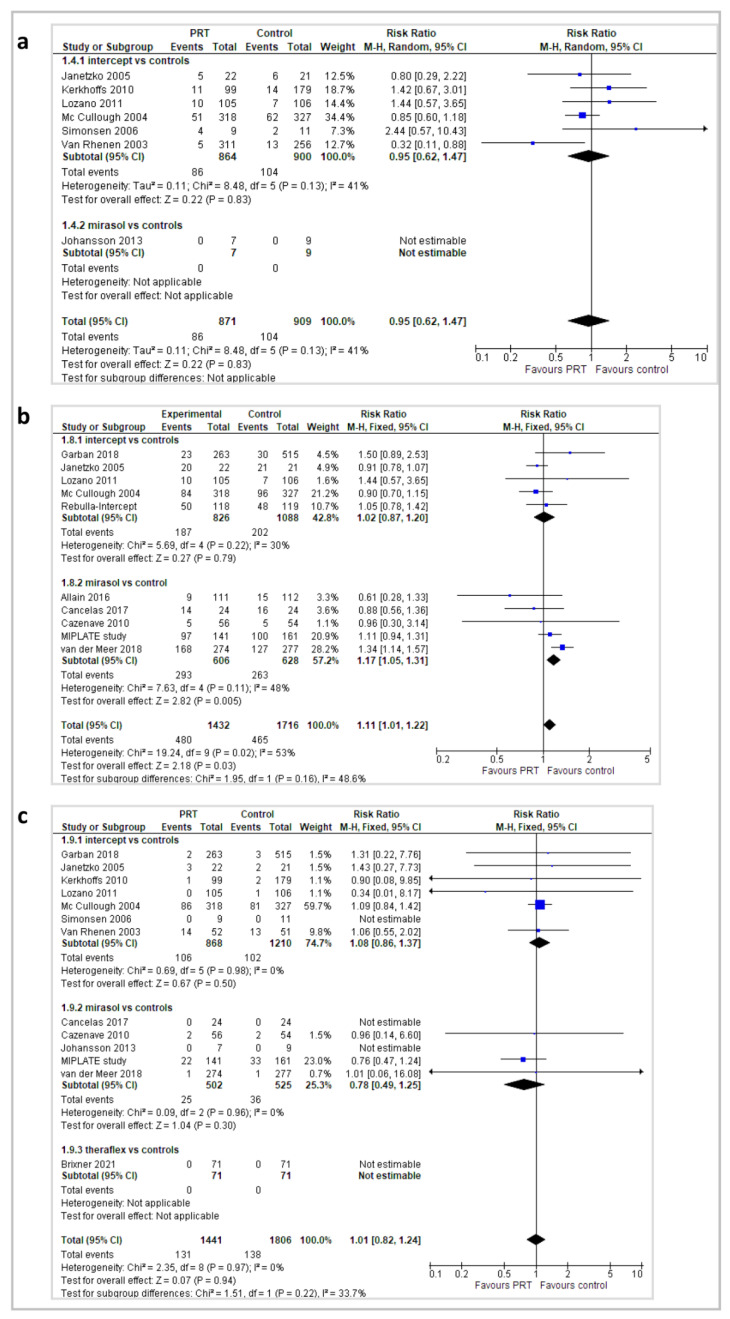
Adverse events: (**a**) number of patients with acute transfusion reactions; (**b**) overall adverse events; (**c**) serious adverse events.

**Figure 5 pathogens-11-00639-f005:**
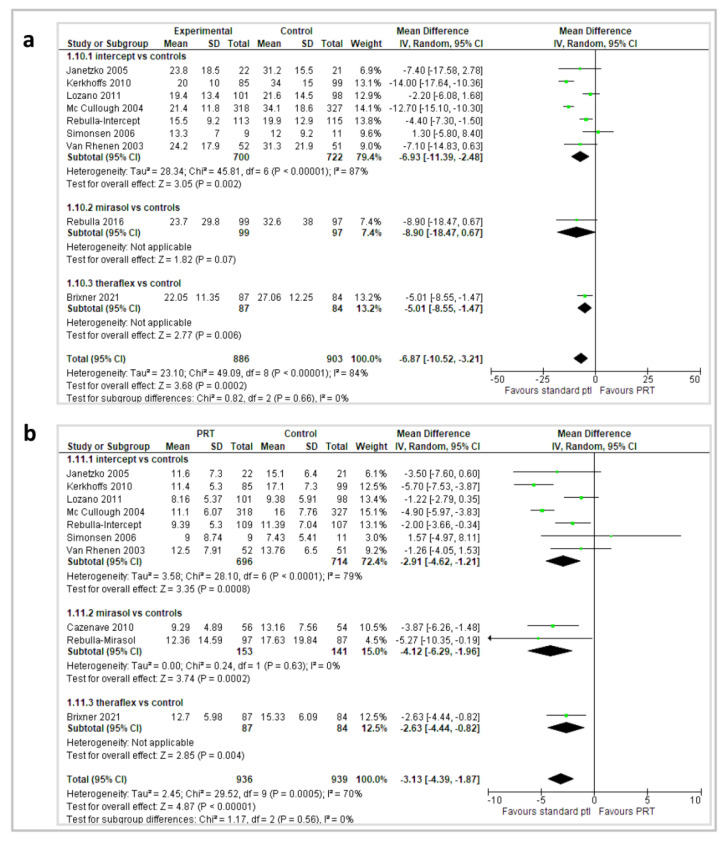
(**a**) 1 h platelet count increment; (**b**) 1 h corrected count increment.

**Figure 6 pathogens-11-00639-f006:**
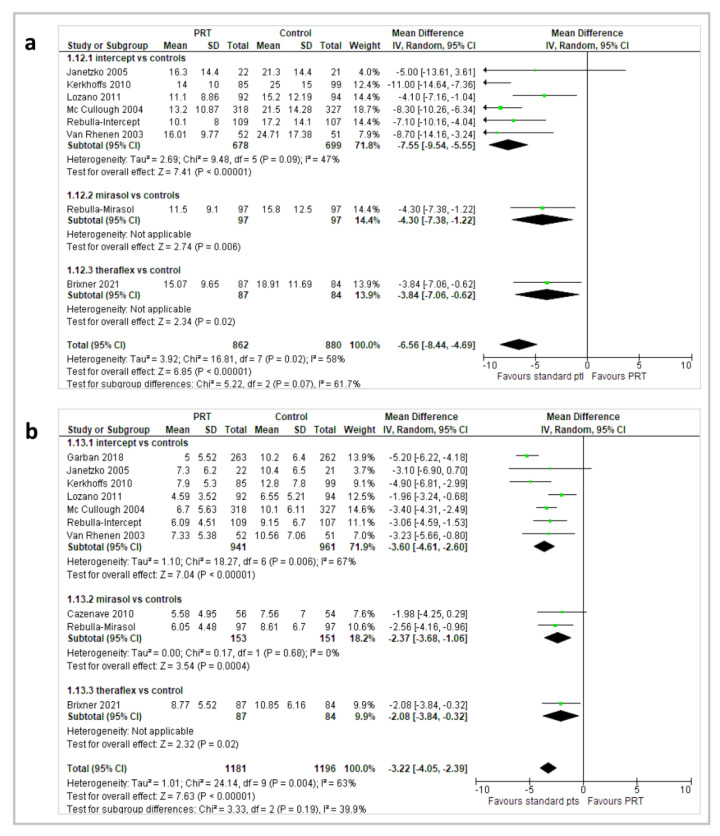
(**a**) 24 h platelet count increment; (**b**) 24 h corrected count increment.

**Figure 7 pathogens-11-00639-f007:**
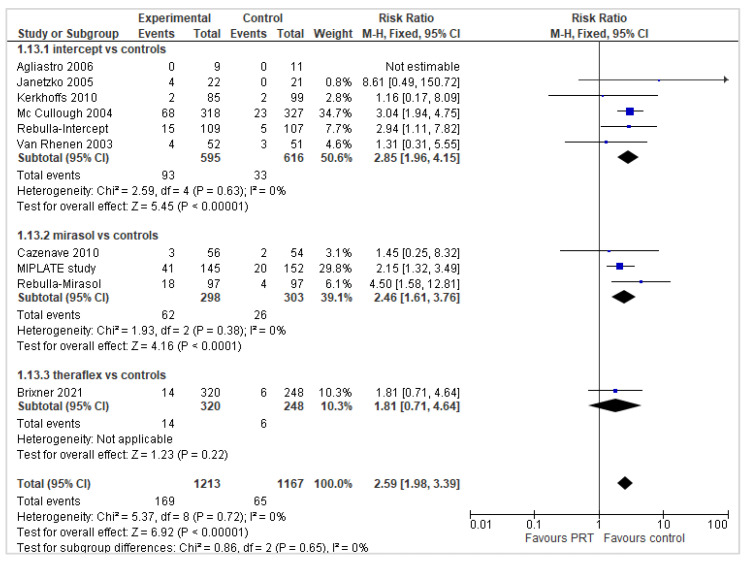
Number of patients with platelet transfusion refractoriness.

**Figure 8 pathogens-11-00639-f008:**
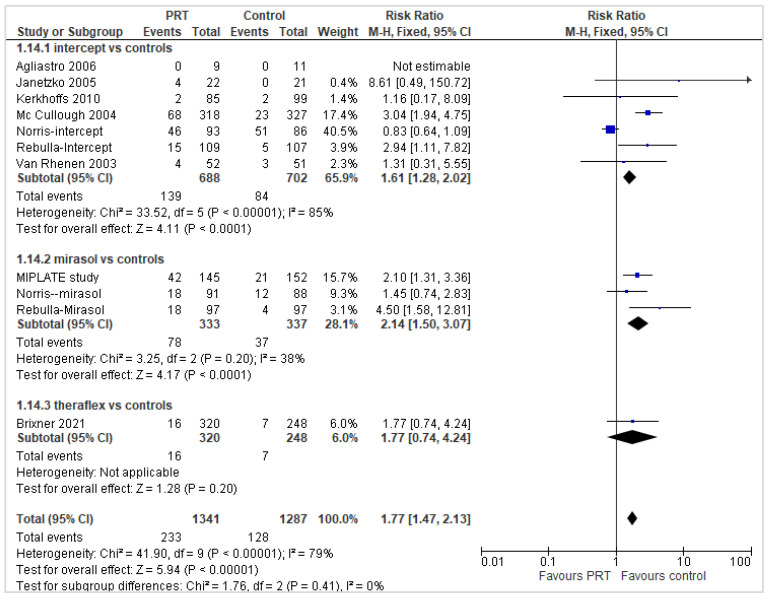
Number of patients with platelet transfusion refractoriness and alloimmunization.

**Figure 9 pathogens-11-00639-f009:**
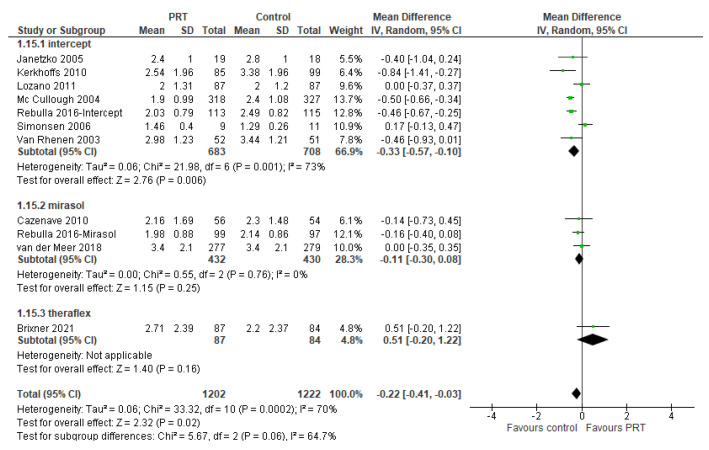
Platelet transfusion interval (days).

**Figure 10 pathogens-11-00639-f010:**
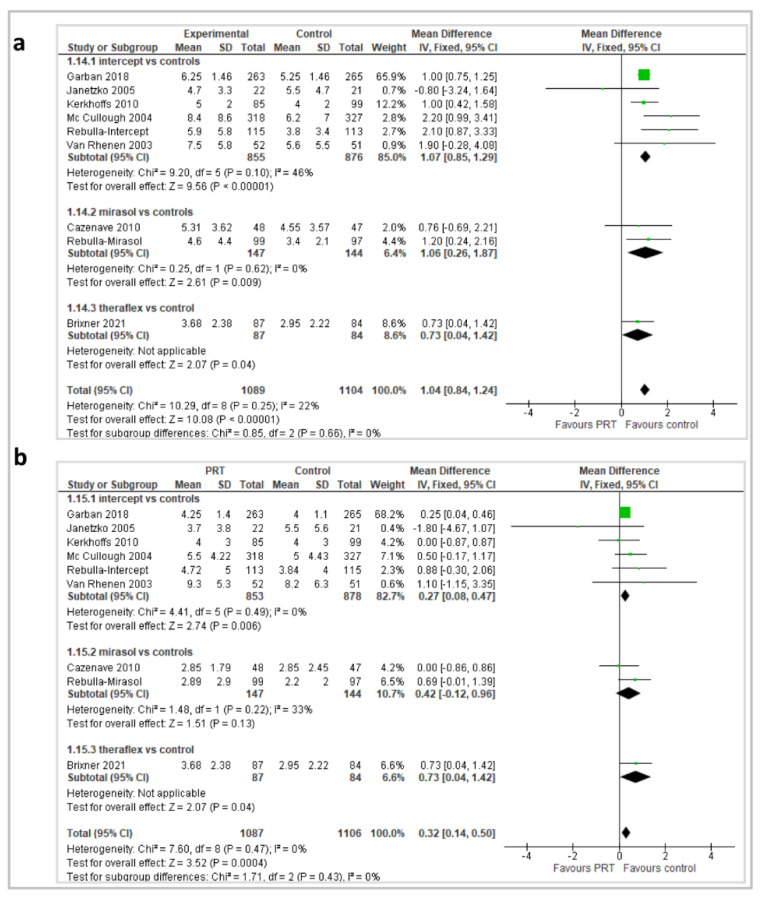
(**a**) Number of platelet transfusions/patient; (**b**) number of RBC transfusions/patient.

**Table 1 pathogens-11-00639-t001:** Characteristics and main results of the included studies on pathogen-reduced platelets.

Study (Year) [Ref]	Study Design	Study Population	Pathogen Reduction Technology	Control	Outcome/s	Main Results
**Kerkhoffs JLH. (2010)** **[16]**	RCT, parallel group	Hemato-oncological pts with thrombocytopenia or expected to be thrombocytopenic caused by myelosuppression. Recruited: 295, treated: 278.	Intercept^®^	St-PLTs	Bleeding assessments, no. of PLT and RBC transfusions, PLT transfusion interval, CI and CCI 1 and 24 h post-transfusion, refractoriness, alloimmunization, adverse transfusion reactions	Pathogen reduction of PLTs probably leads to decreased PLT viability and perhaps compromises hemostatic function.
**MIPLATE study. (2016–2021)** **[17]**	RCT, DB, parallel group	Hemato-oncological pts. Recruited: 422, treated: 330.	Mirasol^®^	St-PLTs	Bleeding assessments, refractoriness, alloimmunization, adverse transfusion reactions	The results show an increase in bleeding and refractoriness in treatment with PR-PLTs compared with control. There are no significant differences for serious adverse events.
**Lozano M. (2011)** **[18]**	RCT, DB, parallel group	Hemato-oncological pts. Recruited: 242, treated: 211.	Intercept^®^	St-PLTs	PLT transfusion interval, CI and CCI 1 and 24 h post-transfusion, adverse transfusion reactions	PR-PLTs stored for up to 7 d provided 1 h CCI and CI within therapeutic ranges not significantly inferior to St-PLTs.
**Janetzko K.** **(2005)** **[19]**	RCT, DB, parallel group	Pts with thrombocytopenia and hemato-oncological diagnosis. Recruited: 43, treated: 43.	Intercept^®^	St-PLTs	Bleeding assessments, no. of PLT and RBC transfusions, PLT transfusion interval, CI and CCI 1 and 24 h post-transfusion, refractoriness, alloimmunization, adverse transfusion reactions	PR-PLT concentrates provide effective PLT transfusion support to thrombocytopenic patients and adequate hemostasis.
**McCullough J. (2004)** **[20]**	RCT, DB, parallel group	Patients with thrombocytopenia. Recruited: 671, treated: 645.	Intercept^®^	St-PLTs	Bleeding assessments, no. of PLT and RBC transfusions, PLT transfusion interval, CI and CCI 1 and 24 h post-transfusion, refractoriness, alloimmunization, adverse transfusion reactions	PR-PLTs were clinically effective in maintaining hemostasis and appear to be associated with an acceptable safety profile.
**Rebulla P. (2017)** **[21]**	RCT, parallel group	Hemato-oncological pts. Recruited: 438, treated: 424 (228 Intercept^®^, 196 Mirasol^®^).	Intercept^®^, Mirasol^®^	St-PLTs	Bleeding assessments, no. of PLT and RBC transfusions, PLT transfusion interval, CI and CCI 1 and 24 h post-transfusion, refractoriness, alloimmunization, adverse transfusion reactions	The study provides additional information on the safety and efficacy of PR-PLTs treated with two commercial pathogen reduction technologies.
**Simonsen AC. (2006)** **[22]**	RCT, DB, parallel group, crossover	Hemato-oncological pts. Recruited: 28, treated: 25.	Intercept^®^	St-PLTs	PLT transfusion interval, CI and CCI 1 h post-transfusion, adverse transfusion reactions	This study failed to show noninferiority within the specified margin of inferiority; 7-day-old PR-PLTs showed acceptable efficacy and safety compared with 7-day-old St-PLTs.
**Van Rhenen D. (2003)** **[23]**	RCT, DB, parallel group	Hemato-oncological pts. Recruited: 103, treated: 103.	Intercept^®^	St-PLTs	Bleeding assessments, no. of PLT and RBC transfusions, PLT transfusion interval, CI and CCI 1 and 24 h post-transfusion, refractoriness, alloimmunization, adverse transfusion reactions	PR-PLTs offer the potential to further improve the safety of PLT transfusion using technology compatible with current methods to prepare buffy coat PLT components.
**Brixner V. (2021)** **[24]**	RCT, DB, parallel group	Hemato-oncological pts. Recruited: 175, treated: 171.	Theraflex^®^	St-PLTs	Bleeding assessments, no. of PLT and RBC transfusions, PLT transfusion interval, CI and CCI 1 and 24 h post-transfusion, refractoriness, alloimmunization, adverse transfusion reactions	Transfusion of PR-PLTs produced with the UVC technology is safe, but noninferiority was not demonstrated.
**Garban F. (2018)** **[25]**	RCT, parallel group	Hemato-oncological pts. Recruited: 842, treated: 795.	Intercept^®^	St-PLTs	Bleeding assessments, no. of PLT and RBC transfusions, CCI 24 h post-transfusion, adverse transfusion reactions	The hemostatic efficacy of PR to PLTs in additive solution; such noninferiority was not achieved when comparing PR-PLTs with PLTs in plasma.
**Van der Meer PF. (2018)** **[26]**	RCT, parallel group	Hemato-oncological pts. Recruited: 469, treated: 469.	Mirasol^®^	St-PLTs	Bleeding assessments, PLT transfusion interval, adverse transfusion reactions	The noninferiority criterion for PR-PLTs was met in the intention-to-treat analysis.
**Johansson PI. (2013)** **[27]**	RCT, parallel group, crossover	Hemato-oncological pts. Recruited: 15, treated: 15.	Mirasol^®^	St-PLTs	Adverse transfusion reactions	PR-PLTs that remain in circulation provide comparable hemostatic function to untreated PLTs.
**Allain JP. (2016)** **[28]**	RCT, DB, parallel group	Hematological pts. Recruited: 227, treated: 223.	Mirasol^®^	Standard whole blood	Adverse transfusion reactions	There was no increase in adverse events in patients who received the treated blood.
**Cancelas JA. (2017)** **[29]**	RCT, parallel group	In vitro 41 subjects, in vivo 26 subjects.	Mirasol^®^	Standard RBCs	Adverse transfusion reactions	RBCs prepared using amustaline pathogen reduction meet the FDA criteria for post-transfusion recovery and are metabolically and physiologically appropriate for transfusion following 35 days of storage.
**Slichter SJ. (2006)** **[30]**	RCT, parallel group, crossover	Hemato-oncological pts. Recruited: 60, treated: 32.	Intercept^®^	St-PLTs	Bleeding assessments, CCI 1 and 24 h post-transfusion, CI and CCI 18 to 24 h, adverse transfusion reactions	PR-PLTs provided correction of prolonged bleeding times and transfusion intervals not significantly different than reference PLTs despite significantly lower PLT count increments and CCIs.
**Cazenave JP. (2010)** **[31]**	RCT, parallel group	Hemato-oncological pts. Recruited: 118, treated: 110.	Mirasol^®^	St-PLTs	Bleeding assessments, no. of PLT and RBC transfusions, PLT transfusion interval, CCI 1 and 24 h post-transfusion, refractoriness, adverse transfusion reactions	The study failed to show noninferiority of PR-PLTs based on predefined CCI criteria.
**Agliastro RE. (2006)** **[32]**	RCT, parallel group	Hemato-oncological pts. Recruited: 30, treated: 30.	Intercept^®^	St-PLTs	Refractoriness, alloimmunization	The study was only available as an abstract and did not provide usable data on all adverse events.
**Norris JP. (2018)** **[33]**	RCT, parallel group	Hemato-oncological pts. Recruited: 358, treated: 358 (179 Intercept^®^, 179 Mirasol^®^).	Intercept^®^, Mirasol^®^	St-PLTs	Alloimmunization	The study was not sufficiently powered to determine whether pathogen reduction treatment provides protection from human leukocyte antigen alloimmunization in PLT transfusion recipients.
**De Francisci G. (2004)** **[34]**	RCT, parallel group	16 children with congenital cyanogen cardiopathy, 28 adults with cirrhosis who are thrombocytopenic. Recruited: 44, treated: 44.	Intercept^®^	St-PLTs	Bleeding assessments, CCI 1 and 24 h post-transfusion, adverse transfusion reactions	Study published as abstract, original study protocol not available for comparison. Minimal participant background characteristics reported. For the cirrhotic group, 1 h CCI not reported. No standard deviations reported for mean 1 and 24 h CCIs. Pre- and post-transfusion PLT counts not reported for either intervention.

RCT: randomized controlled trial; DB: double binding; PLTs: platelets; St-PLTs: standard platelets; PR-PLTs: pathogen-reduced PLTs; CI: platelet count increment; CCI: corrected count increment; pts: patients; no.: number.

**Table 2 pathogens-11-00639-t002:** Summary of findings.

Pathogen Reduction PLTs						
Patient or population: 17 trials in hemato-oncological patients Settings: in and outpatients Intervention: PR-PLTs Comparison: St-PLTs						
Outcomes	Illustrative comparative risks * (95% *CI*)		Relative effect (95% *CI*)	No. of participants (studies)	Quality of the evidence (GRADE)	Comments
	Assumed risk	Corresponding risk				
	Controls (St-PLTs)	Intervention (PR-PLTs)				
Bleeding events						
Any bleeding events	666 per 1000	699 per 1000 (from 559 to 859)	RR 1.03 (95% *CIs*, −0.85 to 1.24)	1931 patients (7 trials, 6 with Intercept^®^, 1 with Mirasol^®^)	⊕⊕⊕⊕ high ^1^	No between-groups difference in the occurrence of bleeding was observed in the overall analysis and in subgroup analyses of Intercept^®^ and Mirasol^®^ trials.
Significant bleeding	390 per 1000	452 per 1000 (from 397 to 514)	RR 1.16 (95% *CIs*, 1.02/1.32)	3033 patients (9 trials, 5 with Intercept^®^, 4 with Mirasol^®^)	⊕⊕⊕⊝ moderate ^	Significant bleeding (WHO grade ≥ 2) was more commonly observed in PR-PLT group compared with St-PLT in the overall analysis, although no between-groups difference was observed in subgroup analysis of Intercept^®^ and Mirasol^®^ trials.
Severe bleeding	54.8 per 1000	59.7 per 1000 (from 41.6 to 85.4)	RR 1.09 (95% *CIs*, 0.76/1.56)	3299 patients (11 trials, 6 with Intercept^®^, 4 with Mirasol^®^, 1 with Theraflex^®^)	⊕⊕⊕⊝ moderate ^	For the outcome severe bleeding (WHO grade ≥ 3), no between-groups difference was observed in the overall analysis and in subgroup analyses of Intercept^®^, Mirasol^®^, and Theraflex^®^ trials.
Adverse events						
Any adverse event	292 per 1000	318 per 1000 (from 294 to 347)	RR 1.09 (95% *CIs*, 1.01/1.19)	3345 patients (11 trials, 6 with Intercept^®^ and 5 with Mirasol^®^)	⊕⊕⊝⊝ low ^2^	In the overall analysis and in the subgroup of Mirasol^®^ trials, overall adverse events were more commonly observed in PR-PLT group compared with St-PLTs. No between-groups difference was observed in subgroup analysis of Intercept^®^ trials.
Serious adverse events	76 per 1000	76 per 1000 (from 62 to 94)	RR 1.01 (95% *CIs*, 10.82/1.24)	3247 patients (11 trials, 7 with Intercept^®^, 4 with Mirasol^®^, 1 with Theraflex^®^)	⊕⊕⊕⊝ moderate ^3^	No between-groups difference in the occurrence of serious adverse events was observed in the overall analysis and in subgroup analyses of Intercept^®^, Mirasol^®^, and Theraflex^®^ trials.
PLT Count increment						
1 h CI	The mean 1 h CI ranged across St-PLT group from 13.2 to 24.2	The mean 1 h CI score in PR group was from 1.3 higher to 14 lower	MD −6.87 (95% *CIs*, −10.52 to −3.21)	1847 patients (10 trials, 8 with Intercept^®^, 1 with Mirasol^®^, and 1 with Theraflex^®^)	⊕⊕⊕⊝ moderate ^	Combining data across 10 trials showed that participants who received PR-PLT transfusions had a lower 1 h CI.
1 h CCI	The mean 1 h CCI ranged across St-PLT group from 7.4 to 17.1	The mean 1 h CCI score in PR group was from 1.57 higher to 5.7 lower	MD −3.13 (95% *CIs*, −4.39 to −1.87)	1933 patients (11 trials, 8 with Intercept^®^, 3 with Mirasol^®^, 1 with Theraflex^®^)	⊕⊕⊕⊝ moderate ^	In the overall analysis and in subgroup analyses, participants who received PR-PLT transfusions had a lower 1 h CCI.
24 h CI	The mean 24 h CI ranged across St-PLT group from 15.8 to 25	The mean 24 h CI score in PR group was from 3.84 to 11 lower	MD −6.65 (95% *CIs*, −8.44 to −4.86)	1800 patients (9 trials, 7 with Intercept^®^, 1 with Mirasol^®^, 1 with Theraflex^®^)	⊕⊕⊕⊝ moderate ^^^	In the overall analysis and in subgroup analyses, participants who received PR-PLT transfusions had a lower 24 h CI.
24 h CCI	The mean 24 h CCI ranged across St-PLT group from 7.5 to 12.8	The mean 24 h CCI score in PR group was from 1.98 to 5.20 lower	MD −3.18 (95% *CIs*, −3.96 to −2.41)	2435 patients (11 trials, 8 with Intercept^®^, 2 with Mirasol^®^, 1 with Theraflex^®^)	⊕⊕⊕⊝ moderate ^	In the overall analysis and in subgroup analyses, participants who received PR-PLT transfusions had a lower 24 h CCI.
Patients with refractoriness						
No. of patients with PLT refractoriness	55.6 per 1000	144 per 1000 (from 110 to 188)	RR 2.59 (95% *CIs*, 1.98/3.39)	2389 patients (10 trials, 6 with Intercept^®^, 3 with Mirasol^®^, 1 with Theraflex^®^	⊕⊕⊕⊕ high ^1^	In the overall analysis and in the subgroup of Intercept^®^ and Mirasol^®^ trials, the no. of patients with PLT refractoriness was significantly higher in PR-PLT group compared to St-PLT. No statistically significant between-groups difference was observed in a single trial with Theraflex^®^.
No. of patients with PLT refractoriness and alloimmunization	99.4 per 1000	175 per 1000 (from 146 to 211)	RR 1.77 (95% *CIs*, 1.47/2.13)	2628 patients (11 trials, 7 with Intercept^®^, 3 with Mirasol^®^, 1 with Theraflex^®^)	⊕⊕⊕⊕ high ^1^	In the overall analysis and in the subgroup of Intercept^®^ and Mirasol^®^ trials, the no. of patients with PLT refractoriness and alloimmunization was significantly higher in PR-PLT group compared with St-PLT. No statistically significant between-groups difference was observed in a single trial with Theraflex^®^.
PLT and RBC transfusions						
No. of PLT transfusions/participants	The mean no. of PLT. transfusions in St-PLT recipients ranged from 2.95 to 6.2	The mean no. of PLT transfusions in PR-PLT recipients ranged from 3.68 to 8.4	MD 1.04 (95% *CIs*, 0.84/1.24)	2194 patients (9 trials, 6 with Intercept^®^, 2 with Mirasol^®^, 1 with Theraflex^®^)	⊕⊕⊕⊕ high ^1^	In the overall analysis and in subgroup analyses, PR-PLT recipients had a higher no. of PLT transfusions compared with St-PLT recipients.
No. of RBC transfusions/participants	The mean no. of PLT transfusions in St-PLT recipients ranged from 2.2 to 5.5	The mean no. of PLT transfusions in PR-PLT recipients ranged from 2.85 to 5.5	MD 0.32 (95% *CIs*, 0.14/0.50)	2193 patients (9 trials, 6 with Intercept^®^, 2 with Mirasol^®^, 1 with Theraflex^®^)	⊕⊕⊕⊕ high ^1^	In the overall analysis and in subgroup analyses of Intercept^®^ and Theraflex^®^ trials, PR-PLT recipients had a higher no. of PLT transfusions compared with St-PLT recipients. No significant between-groups difference was observed in Mirasol^®^ trials.

CI: confidence interval; MD: mean difference; RR: risk ratio. GRADE Working Group grades of evidence. High quality: Further research is very unlikely to change our confidence in the estimate of effect. Moderate quality: Further research is likely to have an important impact on our confidence in the estimate of effect and may change the estimate. Low quality: Further research is very likely to have an important impact on our confidence in the estimate of effect and is likely to change the estimate. Very low quality: We are very uncertain about the estimate. No., number; PLT, platelet; St-PLTs, standard-PLTs; PR-PLT, pathogen-reduced platelets; CI, platelet count increment; CCI, platelet corrected count increment; ROB, risk of bias. ^1^ No need for downgrading was found. ^ Downgraded once for inconsistency (heterogeneity). ^2^ Downgraded twice for ROB (differences in the definition and assessment of overall adverse events) and inconsistency (substantial heterogeneity). ^3^ Downgraded for imprecision because most of the trials were underpowered to detect the occurrence of rare outcomes. * The basis for the assumed risk is the mean control group risk across studies. The corresponding risk (and its 95% confidence interval) is based on the assumed risk in the comparison group and the relative effect of the intervention (and its 95% *CI*).

## Data Availability

Not applicable.

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
