# Peer review of "Efficacy and Safety of Pathogen-Reduced Platelets Compared with Standard Apheresis Platelets: A Systematic Review of RCTs"

_pathogens, 2022, doi:10.3390/pathogens11060639_

Round 1

Reviewer 1 Report

One major comment: in the Table 1 (studies, included in the meta-analysis) you included paper from prof Allain team (#27). That article dedicated to transfusion of pathogen-reduced whole blood, and do not match with the goal of your study. It must be excluded with recalculation of the rest of data.

Author Response

We thank the referee for the review. Below are the responses to the comments.

  1. One major comment: in the Table 1 (studies, included in the meta-analysis) you included paper from prof Allain team (#27). That article dedicated to transfusion of pathogen-reduced whole blood, and do not match with the goal of your study. It must be excluded with recalculation of the rest of data.

Answer: 

Actually, the study by Allain was included in the quantitative analysis just for the evaluation of adverse events of Mirasol technology, but not for the other outcomes related to PRT, and as such we would like to include it in the systematic review. We have now specified this in the results.

Reviewer 2 Report

The review manuscript entitled: "Efficacy and safety of pathogen-reduced platelets compared to standard apheresis platelets: a systematic review of RCTs" by Ilaria Pati et al provides an updated summary of the RCTs conducted on PR-PLTs and the conclusions obtained at this point in time.  The review was well conducted and the summary is well presented and written.  This review is helpful for blood operators considering the implementation of those technologies.

Comments:

Why were only apheresis PLTs considered?

In the abstract, PR-PLT and PRT-PLT should be consistently abbreviated. The sentence on line 20/21 should be reworded as it is not clear.

In the intro, line 72, 'most' is confusing as the listed PRTs are all UV based and therefore ALL not 'most' targeting RNA/DNA. The paragraphs starting with lines 70 and 74 should have some citations on the topics, repsectively. Line 79, 'allows' is not clear and needs a bit more context. The extention of the shelf-life for pr-PLTs is based on risk reduction but comes with the drawback of the arguments listed later in the parahgraph and should be combined and somewhat evaluated. in line 86, 'evaluates' should be changed to 'evaluated'. 

In Figure 1, it is not clear what the difference is between 'duplicates' (second box from top) and 'exclusion' in right box 3rd row.

The figure legend to Figure 2 should have a link to the M&M section describing the judgement assessment. It was not clear to this reviewer in the first place how the assessment was conducted until reading the corresponding M&M section. On this note, it would be helpful if not crucial to distinguish between the review authors' risk assessment and judgements mentioned in the respective manuscripts publishing the RCT data.

Table 2 is very difficult to read as the text in the narrow columns is tough to navigate.

In the discussion, what is ROB (line 342)? In line 352, please change to reviews as they are two.

The tiny conclusion section on page 26 seems to be redundant as there is a nice conclusion paragraph in the discussion section. It could rather be consider to add a few words into this paragraph on what kind of data/info is missing to better understand the impact of PRT in PLTs? Therefore, this paragraph could be Future Perspective.

Author Response

We thank the referee for the review. Below are the responses to the comments.

1. Why were only apheresis PLTs considered?

Answer: The studies included in our meta-analysis focus on apheresis platelets. Compared to plasma, platelets are the major blood component involved in contamination phenomena, also due to the storage temperature, and therefore more frequently subjected to inactivation. Most of the plasma is delivered to the fractionation company for the production of plasma-derived drugs, and inactivated with solvent-detergent treatment.

2. In the abstract, PR-PLT and PRT-PLT should be consistently abbreviated.

Done

3. The sentence on line 20/21 should be reworded as it is not clear.

Answer: It is not clear which sentence the referee is referring to.

4. In the intro, line 72, 'most' is confusing as the listed PRTs are all UV based and therefore ALL not 'most' targeting RNA/DNA.

Done

5. The paragraphs starting with lines 70 and 74 should have some citations on the topics, respectively.

Done

6. Line 79, 'allows' is not clear and needs a bit more context.

Answer: As reported in the EDQM Guide to the preparation, use and quality assurance of blood components, “The maximum storage time for Platelets, Rec, SU is 5 days. Storage may be extended to 7 days, in conjunction with appropriate detection or reduction of bacterial contamination”. The reference has been entered in the text.

7. In line 86, 'evaluates' should be changed to 'evaluated'.

Done

8. In Figure 1, it is not clear what the difference is between 'duplicates' (second box from top) and 'exclusion' in right box 3rd row.

Answer: “Duplicates” means that there was more than one record reporting the same information retrieved concurrently. This is one of the common reasons of exclusion of a record from the systematic review/meta-analysis.

9. The figure legend to Figure 2 should have a link to the M&M section describing the judgement assessment. It was not clear to this reviewer in the first place how the assessment was conducted until reading the corresponding M&M section. On this note, it would be helpful if not crucial to distinguish between the review authors' risk assessment and judgements mentioned in the respective manuscripts publishing the RCT data.

Answer: A link to the M&M section has been added to fig.2. As specified in the M&M, the risk of bias assessment  in the current meta-analysis was conducted according to the domain-based evaluation described in the Cochrane Handbook for Systematic Reviews of Interventions, basing on information reported in the primary studies.

10. Table 2 is very difficult to read as the text in the narrow columns is tough to navigate.

Answer: In this regard, we ask the Editor if it is possible to improve the formatting of the table if the manuscript were to be published.

11. In the discussion, what is ROB (line 342)?

Answer: Risk of Bias (ROB). We proceded to write in full in the text.

12. In line 352, please change to reviews as they are two.

Done

13. The tiny conclusion section on page 26 seems to be redundant as there is a nice conclusion paragraph in the discussion section. It could rather be consider to add a few words into this paragraph on what kind of data/info is missing to better understand the impact of PRT in PLTs? Therefore, this paragraph could be Future Perspective.

Answer: We modified and expanded this section as suggested by the reviewer.

Reviewer 3 Report

The review entitled "Efficacy and safety of pathogen-reduced platelets compared to 2 standard apheresis platelets: a systematic review of RCTs" by Ilaria Pati et al.

Presents a systematic, clearly presented and well-organized review that covers and evaluates the efficacy and safety of blood components treated with pathogen reduction technologies.

The methods used to elaborate the review are comprehensive and the analysis is well-done. The data coming from Italian sources it is a bit dominant on the paper. More general data worldwide would be appreciated if possible.

Author Response

We thank the referee for the review provided. Below are the responses to the referee's comments.

  1. The data coming from Italian sources it is a bit dominant on the paper. More general data worldwide would be appreciated if possible.

Answer: we have performed the bibliographic search without language and geographical limitations, and the records we have found and included in the systematic review reflect what is currently available from the literature. With the exception of one trial from Ghana, studies were conducted in industrialised countries, including USA, Canada, and Europe (France, Germany, Belgium, the Netherlands, Sweden, Spain, and Italy). We have added a sentence related to the geographical distribution of trials in the results.